# Anti-HER2 Cancer-Specific mAb, H_2_Mab-250-hG_1_, Possesses Higher Complement-Dependent Cytotoxicity than Trastuzumab

**DOI:** 10.3390/ijms25158386

**Published:** 2024-08-01

**Authors:** Hiroyuki Suzuki, Tomokazu Ohishi, Tomohiro Tanaka, Mika K. Kaneko, Yukinari Kato

**Affiliations:** 1Department of Antibody Drug Development, Tohoku University Graduate School of Medicine, 2-1 Seiryo-machi, Aoba-ku, Sendai 980-8575, Japan; hiroyuki.suzuki.b4@tohoku.ac.jp (H.S.); tomohiro.tanaka.b5@tohoku.ac.jp (T.T.); mika.kaneko.d4@tohoku.ac.jp (M.K.K.); 2Institute of Microbial Chemistry (BIKAKEN), Numazu, Microbial Chemistry Research Foundation, 18-24 Miyamoto, Numazu-shi, Shizuoka 410-0301, Japan; ohishit@bikaken.or.jp; 3Institute of Microbial Chemistry (BIKAKEN), Laboratory of Oncology, Microbial Chemistry Research Foundation, 3-14-23 Kamiosaki, Shinagawa-ku, Tokyo 141-0021, Japan

**Keywords:** HER2, cancer-specific monoclonal antibody, antitumor effect, complement-dependent cellular cytotoxicity

## Abstract

Cancer-specific monoclonal antibodies (CasMabs) that recognize cancer-specific antigens with in vivo antitumor efficacy are innovative therapeutic strategies for minimizing adverse effects. We previously established a cancer-specific anti-human epidermal growth factor receptor 2 (HER2) monoclonal antibody (mAb), H_2_Mab-250/H_2_CasMab-2. In flow cytometry and immunohistochemistry, H_2_Mab-250 reacted with HER2-positive breast cancer cells but did not show reactivity to normal epithelial cells. In contrast, a clinically approved anti-HER2 mAb, trastuzumab, strongly recognizes both breast cancer and normal epithelial cells in flow cytometry. The human IgG_1_ version of H_2_Mab-250 (H_2_Mab-250-hG_1_) possesses compatible in vivo antitumor effects against breast cancer xenografts to trastuzumab despite the lower affinity and effector activation than trastuzumab in vitro. This study compared the antibody-dependent cellular cytotoxicity (ADCC) and complement-dependent cellular cytotoxicity (CDC) between H_2_Mab-250-hG_1_ and trastuzumab. Both H_2_Mab-250-hG_1_ and trastuzumab showed ADCC activity against HER2-overexpressed Chinese hamster ovary -K1 and breast cancer cell lines (BT-474 and SK-BR-3) in the presence of human natural killer cells. Some tendency was observed where trastuzumab showed a more significant ADCC effect compared to H_2_Mab-250-hG_1_. Importantly, H_2_Mab-250-hG_1_ exhibited superior CDC activity in these cells compared to trastuzumab. Similar results were obtained in the mouse IgG_2a_ types of both H_2_Mab-250 and trastuzumab. These results suggest the different contributions of ADCC and CDC activities to the antitumor effects of H_2_Mab-250-hG_1_ and trastuzumab, and indicate a future direction for the clinical development of H_2_Mab-250-hG_1_ against HER2-positive tumors.

## 1. Introduction

Human epidermal growth factor receptor 2 (HER2) is a member of the receptor tyrosine kinases. Heterodimerization of HER2 with other HER family members and the ligands or ligand-independent HER2 homodimerization results in the autophosphorylation of the cytoplasmic domain. This event initiates a variety of signaling, such as the RAS and PI3K pathways, leading to cancer cell proliferation, survival, and invasiveness [1]. The overexpression of HER2 is observed in approximately 20% of breast cancers [2] and 20% of gastric cancers [3], which are associated with higher rates of recurrence and shorter overall survival.

Trastuzumab, an anti-HER2 monoclonal antibody (mAb), exhibited in vitro anti-proliferative efficacy and a potent antitumor effect in vivo [4,5]. The combination of chemotherapy with trastuzumab in HER2-positive breast cancer patients with metastasis improves the progression-free survival and overall survival [6]. Trastuzumab was approved by the U.S. Food and Drug Administration (FDA) for the treatment of HER2-positive breast cancer [6] and has been the most effective therapy for it for more than 20 years [7]. Trastuzumab is administered in patients with HER2-overexpressed tumors, which are defined by solid and complete membranous staining of more than 10% of cells in immunohistochemistry (IHC 3+) and/or in situ hybridization (ISH)-amplified [8]. Furthermore, trastuzumab–deruxtecan (T-DXd), a trastuzumab-based antibody–drug conjugate (ADC), has been developed and approved by the FDA [9]. T-DXd exhibited superior efficacy in not only HER2-positive breast cancers [10,11] but also HER2-low (IHC 1+ or IHC 2+/ISH-non-amplified) advanced breast cancers [12] and HER2-mutant lung cancers [13]. Because half of all breast cancers are classifiable as HER2-low, a significant number of patients are estimated to receive the benefit from T-DXd therapy [14].

The immunologic engagement of trastuzumab mediates the clinical efficacy [4]. Antibody-dependent cellular cytotoxicity (ADCC) is elicited by natural killer (NK) cells or macrophages upon the binding of Fcγ receptors (FcγRs) to the Fc region of mAbs [4]. Trastuzumab is a humanized IgG_1_ mAb that binds to FcγRs [15] and activates macrophages, neutrophils, and dendritic cells, which change the adaptive immunity by cytokine production, chemotaxis, and antigen presentation [4]. Moreover, the FcγR binding results in the activation of NK cells and macrophages, which can result in the target cell killing [4]. However, the ADCC is impaired by the *N*-linked glycosylation in the Fc region [16]. In particular, a lack of core fucose on the Fc *N*-glycan enhances the Fc binding to the FcγRs on effector cells [17]. Therefore, a core fucose deficiency on the Fc *N*-glycan has been shown to enhance the binding to FcγR on effector cells [17] and exert potent antitumor effects [18]. The defucosylated recombinant mAbs can be produced using fucosyltransferase 8-knockout Chinese hamster ovary (CHO) cells [19].

Complement-dependent cellular cytotoxicity (CDC) is also exerted by the Fc domain of mAbs [20,21]. Although complements have been thought of as an adjunctive component of the antibody-mediated cytolytic effects, complement is currently considered an essential effector of the tumor cytotoxic responses of mAb-based immunotherapy [21]. Through the development of a chimeric anti-CD20 mAb, rituximab, for the treatment of B cell lymphomas, the involvement with the cytolytic capacity of the complement was revealed in the antitumor effect [22,23]. In not only anti-CD20 but also anti-CD38 and CD52 immunotherapies, the cytolytic capacity of the tumor by complements has been shown [23,24,25]. Furthermore, a growing body of evidence suggests that complements play crucial functions in not only tumor cytolysis but also several immunologic roles in antitumor immunity [26,27]. The crosstalk of complement effectors and cellular signaling pathways influence the T and B cell responses, T helper/effector T cell survival, differentiation, and B cell activation.

A common adverse effect of anti-HER2 mAbs and the ADCs is cardiotoxicity [28]. Routine cardiac monitoring is required for patients [29]. Moreover, the lack of cardiac trabeculae is observed in *ErbB2* (ortholog of *HER2*)-knockout mice [30], and the features of dilated cardiomyopathy are observed in ventricular-specific *ErbB2*-knockout mice [31]. These results indicate that HER2 is involved in normal heart development and homeostasis. Therefore, more selective or specific anti-HER2 mAbs against tumors are required to reduce heart failures.

We previously developed cancer-specific anti-HER2 mAbs, H_2_Mab-214/H_2_CasMab-1 [32] and H_2_Mab-250/H_2_CasMab-2 [33], from 278 clones of anti-HER2 mAbs using glioblastoma LN229-expressed HER2 as an antigen. Notably, both H_2_Mab-214 and H_2_Mab-250 did not react with spontaneously immortalized normal epithelial cells (HaCaT and MCF 10A) [32,33]. Moreover, H_2_Mab-250 did not react with immortalized normal epithelial cells derived from the mammary gland, lung bronchus, gingiva, kidney proximal tubule, thymus, corneal, and colon [33]. In contrast, most anti-HER2 mAbs, including trastuzumab, reacted with both cancer and normal epithelial cells [34]. The epitope mapping revealed that the Trp614 in the HER2 extracellular domain (ECD) 4 mainly contributes to the recognition by H_2_Mab-250 [33]. H_2_Mab-214 also recognized a similar epitope of H_2_Mab-250, and the crystal structure suggests that H_2_Mab-214 recognizes a structurally misfolded region in the HER2-ECD4, which usually forms a β-sheet [32]. The result indicates that the local misfolding in the Cys-rich-ECD4 governs the cancer-specificity of H_2_Mab-214. Furthermore, we produced mouse IgG_2a_-type and human IgG_1_-type mAbs from H_2_Mab-214 and H_2_Mab-250. We found that both H_2_Mab-214 and H_2_Mab-250 possess a compatible in vivo antitumor effect against breast cancer xenografts with trastuzumab despite the lower affinity and effector activation than trastuzumab in vitro [32,34].

This study compared the ADCC and CDC between H_2_Mab-250 and trastuzumab against HER2-overexpressed CHO-K1 (CHO/HER2) and breast cancer cell lines.

## 2. Results

### 2.1. ADCC and CDC of H_2_Mab-250 and Trastuzumab against Breast Cancers

H_2_Mab-250 recognized HER2-positive breast cancers (BT-474 and SK-BR-3) but did not recognize HER2 in normal epithelial cells. In contrast, trastuzumab recognized both types of HER2 [34]. Because H_2_Mab-250 and trastuzumab are mouse IgG_1_ and human IgG_1_, respectively, we produced the human IgG_1_ type of recombinant H_2_Mab-250 (H_2_Mab-250-hG_1_) and trastuzumab and confirmed the reactivity against breast cancers and normal epithelial cells (Appendix A). Although H_2_Mab-250 possesses ~10-fold lower affinity than trastuzumab, H_2_Mab-250-hG_1_ possesses compatible antitumor effects against BT-474 and SK-BR-3 xenografts compared to trastuzumab [34]. To reveal the contribution of the ADCC and CDC activities to the antitumor effects, we performed an in vitro ADCC and CDC assay. Both BT-474 and SK-BR-3 cells were labeled with Calcein-AM. Then, the calcein release was measured due to the cytotoxicity of mAbs plus human NK cell (ADCC) or mAbs plus complements (CDC). As shown in Figure 1A, both H_2_Mab-250-hG_1_ and trastuzumab induced ADCC against BT-474 cells (47% and 64% cytotoxicity, respectively) more effectively than the control human IgG (20% cytotoxicity; *p* < 0.01). Furthermore, trastuzumab showed a superior ADCC compared to H_2_Mab-250-hG_1_ (*p* < 0.01). In contrast, H_2_Mab-250-hG_1_ showed a significant CDC (47% cytotoxicity) compared to the control human IgG (20% cytotoxicity; *p* < 0.05). However, trastuzumab did not show a significant CDC (Figure 1B). In SK-BR-3, both H_2_Mab-250-hG_1_ and trastuzumab induced ADCC (12% and 17% cytotoxicity, respectively) more effectively than the control human IgG (4% cytotoxicity; *p* < 0.05 and *p* < 0.01, respectively, Figure 1C). Moreover, both H_2_Mab-250-hG_1_ and trastuzumab exhibited CDC (42% and 26% cytotoxicity, respectively) more effectively than the control human IgG (6% cytotoxicity; *p* < 0.01, Figure 1D). Importantly, H_2_Mab-250-hG_1_ showed a superior CDC compared to trastuzumab (*p* < 0.01, Figure 1D). These results suggest the different contributions of ADCC and CDC to the antitumor effects of H_2_Mab-250-hG_1_ and trastuzumab.

### 2.2. ADCC and CDC by H_2_Mab-250 and Trastuzumab against CHO/HER2

To confirm the requirement of HER2 in the ADCC and CDC of H_2_Mab-250 and trastuzumab, we used CHO-K1 and CHO/HER2 and performed the ADCC and CDC assays. CHO/HER2 was also recognized by H_2_Mab-250 and trastuzumab with low and high reactivity, respectively [34]. As shown in Figure 2A, both H_2_Mab-250-hG_1_ and trastuzumab induced ADCC against CHO/HER2 cells (70% and 77% cytotoxicity, respectively) more effectively than the control human IgG (13% cytotoxicity; *p* < 0.01). In contrast, H_2_Mab-250-hG_1_ showed a significant CDC (63% cytotoxicity) compared to the control human IgG (10% cytotoxicity; *p* < 0.01, Figure 2B). However, trastuzumab did not show a significant CDC (Figure 2B). In CHO-K1, we did not observe ADCC and CDC in the presence of H_2_Mab-250 and trastuzumab (Figure 2C,D). These results indicate that the recognition of HER2 is essential to the exertion of ADCC and CDC by H_2_Mab-250-hG_1_ and trastuzumab. Furthermore, dose-dependent activation of CDC activity by H_2_Mab-250 and trastuzumab was observed against CHO/HER2, but not CHO-K1 (Figure 2E). H_2_Mab-250-hG_1_ exhibited a significant CDC compared to the control human IgG from 25 µg/mL and showed it compared to trastuzumab at 100 µg/mL (Figure 2E).

### 2.3. Antitumor Activities by H_2_Mab-250-hG_1_ and Trastuzumab

Next, we examined the in vivo antitumor efficacy of H_2_Mab-250-hG_1_ and trastuzumab in the CHO/HER2 xenograft model. We injected H_2_Mab-250-hG_1_, trastuzumab, and control human IgG intraperitoneally on days 7, 14, and 21 after inoculating CHO/HER2. Furthermore, human NK cells were injected around the tumors on the same days of the Abs injection. Following the inoculation, we measured the tumor volume on days 7, 14, 21, and 28. The H_2_Mab-250-hG_1_ and trastuzumab administration led to a significant and similar reduction in the CHO/HER2 xenograft on day 28 (*p* < 0.01) compared with that of the control (Figure 3A). Both H_2_Mab-250-hG_1_ and trastuzumab administration resulted in an 81% reduction in the CHO/HER2 xenograft volume compared with the control human IgG on day 28.

The CHO/HER2 xenografts from the H_2_Mab-250-hG_1_- and trastuzumab-treated mice weighed significantly less than those from the control human IgG-treated mice (93% and 94% reduction, respectively; *p* < 0.05, Figure 3B,C). There was no significant difference between the H_2_Mab-250-hG_1_- and trastuzumab-treated xenografts.

A body weight loss was not observed in the H_2_Mab-250-hG_1_- and trastuzumab-treated CHO/HER2 xenograft-bearing mice (Figure 3D), and there was no difference in the body appearance in those mice (Figure 3E).

We also investigated the pharmacokinetics of H_2_Mab-250-hG_1_ and trastuzumab after administration in nude mice. As shown in Appendix A, the half-lives of H_2_Mab-250-hG_1_ and trastuzumab were determined as 128 and 133 h, respectively. These results indicate that H_2_Mab-250-hG_1_ possesses a similar half-life compared to trastuzumab.

### 2.4. ADCC and CDC by Mouse IgG_2a_-Type H_2_Mab-250 and Trastuzumab against CHO/HER2

We next investigated the ADCC and CDC against CHO-K1 and CHO/HER2 using mouse IgG_2a_-type H_2_Mab-250 (H_2_Mab-250-mG_2a_) and trastuzumab (tras-mG_2a_) to assess the influence of the antibody format. As shown in Figure 4A, tras-mG_2a_ induced ADCC against CHO/HER2 cells (43% cytotoxicity) more effectively than the control mouse IgG_2a_ (10% cytotoxicity; *p* < 0.05). In contrast, H_2_Mab-250-mG_2a_ did not significantly induce ADCC against CHO/HER2 cells (Figure 4A). Furthermore, both H_2_Mab-250-mG_2a_ and tras-mG_2a_ induced CDC against CHO/HER2 cells (51% and 45% cytotoxicity, respectively) more effectively than the control mouse IgG_2a_ (Figure 4B, 27% cytotoxicity; *p* < 0.01 and *p* < 0.05 respectively). In CHO-K1, we did not observe ADCC and CDC in the presence of H_2_Mab-250-mG_2a_ and tras-mG_2a_ (Figure 4C,D).

These results indicate that tras-mG_2a_ showed a superior ADCC compared to H_2_Mab-250-mG_2a_. In contrast, H_2_Mab-250-mG_2a_ exhibited a superior CDC to CHO/HER2 compared to tras-mG_2a._

### 2.5. Antitumor Activities by Mouse IgG_2a_-Type H_2_Mab-250 and Trastuzumab

Next, we examined the in vivo antitumor efficacy of H_2_Mab-250-mG_2a_ and tras-mG_2a_ in the CHO/HER2 xenograft model. We injected H_2_Mab-250-mG_2a_, tras-mG_2a_, and a control mouse IgG_2a_ intraperitoneally on days 9 and 16 after inoculating CHO/HER2. Following the inoculation, we measured the tumor volume on days 9, 16, and 21. The H_2_Mab-250-mG_2a_ and tras-mG_2a_ administration led to a potent and similar reduction in the CHO/HER2 xenograft on days 16 and 21 (*p* < 0.01) compared with that of the control mouse IgG_2a_ (Figure 5A). The H_2_Mab-250-mG_2a_ and tras-mG_2a_ administration resulted in a 77% and 74% reduction in the CHO/HER2 xenograft volume compared with the control mouse IgG_2a_ on day 21.

The CHO/HER2 xenografts from the H_2_Mab-250-mG_2a_- and tras-mG_2a_-treated mice weighed significantly less than those from the control mouse IgG_2a_-treated mice (94% and 94% reduction, respectively; *p* < 0.01, Figure 5B,C). There was no significant difference between the H_2_Mab-250-mG_2a_- and tras-mG_2a_-treated xenografts.

Figure 5D shows that body weight loss was not observed in the H_2_Mab-250-mG_2a_- and tras-mG_2a_-treated CHO/HER2 xenograft-bearing mice. However, a slight difference was observed between the control and H_2_Mab-250-mG_2a_-treated mice. Those mice had no difference in body appearance (Figure 5E).

### 2.6. Comparison of Antitumor Activities by H_2_Mab-250-hG_1_ and Trastuzumab in the Absence of Human NK Cells

As shown in Figure 3, we injected human NK cells with H_2_Mab-250-hG_1_ and trastuzumab because high ADCC activity was expected. Since H_2_Mab-250-hG_1_ possesses a higher CDC activity, we next compared the antitumor effects of H_2_Mab-250-hG_1_ and trastuzumab without the human NK cells. We injected H_2_Mab-250-hG_1_ and trastuzumab intraperitoneally on days 7, 14, and 21 after inoculating CHO/HER2. Following the inoculation, we measured the tumor volume on days 7, 10, 14, 16, 21, 24, and 29. We observed the more potent antitumor efficacy of H_2_Mab-250-hG_1_ in relation to both the tumor volume (*p* < 0.01, Figure 6A) and weight (*p* < 0.01, Figure 6B) compared with that of trastuzumab at day 29 (Figure 6C).

Body weight loss was not observed in the H_2_Mab-250-hG_1_- and trastuzumab-treated CHO/HER2 xenograft-bearing mice (Figure 6D), and there was no difference in body appearance in those mice (Figure 6E).

## 3. Discussion

We have developed CasMabs against HER2 (H_2_Mab-250 [33,34]), podocalyxin (PcMab-6 [35]), and podoplanin (LpMab-2 [36] and LpMab-23 [37]) by evaluating the reactivity against cancer and normal cells in flow cytometry and immunohistochemistry. We also showed the in vivo antitumor effect of the recombinant mAbs (mouse IgG_2a_ or human IgG_1_ types) derived from the abovementioned mAbs [32,33,34]. Especially, H_2_Mab-250 showed a potent antitumor effect in vivo [34] despite the lower reactivity and affinity than trastuzumab in vitro [33]. However, the reason for this has not been clarified. In this study, we compared the ADCC and CDC activity of H_2_Mab-250 and trastuzumab and found that H_2_Mab-250 exhibited a superior CDC activity against breast cancer and HER2-overexpressed cells compared to trastuzumab (Figure 1, Figure 2 and Figure 4). Furthermore, both H_2_Mab-250-hG_1_ and H_2_Mab-250-mG_2a_ showed compatible antitumor effects compared to the corresponding isotype of trastuzumab (Figure 3 and Figure 5). These results suggest that the CDC activity of H_2_Mab-250 would compensate for the lower ADCC activity in the antitumor efficacy.

Complement is an essential effector of tumor cytotoxic responses in mAb-based immunotherapy [21]. The engagement of the Fc domain of mAbs with complement C1q triggers the assembly of the active C1 complex (C1q, C1r, and C1s), which initiates the cascade. The downstream activation of terminal complement components results in the assembly of the pore-forming membrane attack complex (MAC or C5b–C9) on the tumor cell membrane, which promotes the terminal lytic pathway [21]. Complement activation also leads to tumor cell opsonization by C3-derived opsonins (C3b, iC3b, and C3dg), which bind to the CR3/CR4 complement receptors on phagocytes (neutrophils and macrophages) and augment the FcγR-dependent phagocytic uptake of opsonized tumor cells. Furthermore, complement activation generates pro-inflammatory mediators (C3a and C5a). The anaphylatoxin C5a upregulates the FcγRs on phagocytes and primes them for enhanced phagocytosis and increasing the magnitude of the tumor cytolytic response [21].

Because H_2_Mab-250 showed increased CDC activity only in the presence of complement (Figure 1, Figure 2 and Figure 4), the assembly of MAC is thought to be efficiently formed on the cells. Furthermore, the predisposition to CDC of H_2_Mab-250 is independent of the isotype or species of mAbs (Figure 2 and Figure 4). Therefore, the complementarity-determining region and epitope of H_2_Mab-250 are thought to be necessary. In Figure 6, the antitumor effects of H_2_Mab-250-hG_1_ were higher than those of trastuzumab without human NK cells, indicating that H_2_Mab-250-hG_1_ exerts antitumor activities with much higher CDC than trastuzumab in vivo. Several factors, including the antigen size and density, determine the engagement of the classical complement pathway. In addition, a geometry of the antigen–mAb complex allows efficient C1q binding [38]. Furthermore, IgG antibodies can form ordered hexamers upon binding to their antigen on cell surfaces. These hexamers efficiently bind the hexavalent complement component C1q, the first step in the classical pathway of complement activation [39,40]. The structure of the H_2_Mab-250-HER2 complex may provide adequate access for complements to exert CDC. Further investigation and confirmation are required to clarify the mechanisms of CDC in H_2_Mab-250.

Trastuzumab exerts antitumor activity through multiple mechanisms of action but is incapable of eliciting CDC in HER2-positive cancers in the presence of human serum [41,42]. As shown in Figure 1B, trastuzumab did not elicit CDC against BT-474 cells. An anti-HER2 bispecific and biparatopic antibody, zanidatamab, elicited potent CDC against HER2-high tumor cells, including BT-474 cells. Zanidatamab possesses an anti-HER2-ECD4 single-chain variable fragment (scFv) linked to heavy chain 1 and an anti-HER2-ECD2 fragment antigen-binding (Fab) domain on heavy chain 2. Zanidatamab binds adjacent HER2 molecules in trans and initiates distinct HER2 reorganization and large HER2 clusters, which are not observed with trastuzumab [43]. Optimal CDC activity requires hexameric clustering of mAb Fc domains in the mAb–antigen clusters [44]. We identified the epitope of H_2_Mab-250 as _613-_IWKFP_-617_ in the HER2-ECD4. The epitope of trastuzumab is a broader sequence (residues 579-625), which includes the H_2_Mab-250 epitope [33]. It is worthwhile to investigate the ability of H_2_Mab-250 to form a cluster with HER2.

Chimeric antigen receptor (CAR)-T cell therapy is rapidly advancing as a cancer treatment; however, designing an optimal CAR remains challenging. Due to the specific reactivity against cancer cells, H_2_Mab-250 is clinically developed as CAR-T cell therapy, which is evaluated in a phase I study for HER2-positive advanced solid tumors in the US (NCT06241456). We discussed the benefit of reducing CAR affinity to limit trogocytosis, which is observed in the high affinity of CAR-T cells [34]. In the monotherapy of H_2_Mab-250, we have reported compatible antitumor effects against breast cancer xenograft compared to trastuzumab [32,34] and showed the importance of CDC in this study. Extensive research indicates that resistance to CDC is induced by the expression of complement regulators in tumor cells during the escape from host immune responses. Notably, an upregulation of the regulators, including CD46, CD55, and CD59, has been shown to prevent CDC through suppression of terminal complement activation and MAC assembly [45,46,47]. In this regard, several strategies have been developed to overcome the resistance to CDC in mAb-based immunotherapy [48,49,50,51]. Therefore, dual targeting of HER2 by H_2_Mab-250 and complement regulators should be investigated in future studies in in vitro models.

## 4. Materials and Methods

### 4.1. Cell Culture

Chinese hamster ovary (CHO)-K1 cells were obtained from the American Type Culture Collection (ATCC, Manassas, VA, USA). CHO-K1 and HER2-overexpressed CHO-K1 (CHO/HER2) [33] were cultured in the Roswell Park Memorial Institute (RPMI)-1640 medium (Nacalai Tesque, Inc., Kyoto, Japan). BT-474 and SK-BR-3 were also obtained from the ATCC, and they were cultured in Dulbecco’s Modified Eagle Medium (DMEM) (Nacalai Tesque, Inc.). These media were supplemented with 10% fetal bovine serum (FBS; Thermo Fisher Scientific Inc., Waltham, MA, USA), antibiotic–antimycotic mixed solution (Nacalai Tesque, Inc.). All the cell lines were cultured at 37 °C in a humidified atmosphere with 5% CO_2_ and 95% air.

### 4.2. Production of Recombinant mAbs

To generate H_2_Mab-250-hG_1_, V_H_ of H_2_Mab-250 and C_H_ of human IgG_1_ were cloned into the pCAG-Ble vector. The V_L_ of the H_2_Mab-250 and C_L_ of the human kappa light chain were cloned into the pCAG-Neo vector.

To generate H_2_Mab-250-mG_2a_, we cloned the V_H_ cDNA of H_2_Mab-250 and C_H_ of mouse IgG_2a_ into the pCAG-Ble vector. To generate a mouse IgG_2a_ type of trastuzumab (tras-mG_2a_), the V_H_ cDNA of trastuzumab and the C_H_ cDNA of mouse IgG_2a_ were cloned into the pCAG-Neo vector, and the V_L_ cDNA of trastuzumab and the C_L_ cDNA of mouse kappa light chain were cloned into the pCAG-Ble vector.

To generate the control mouse IgG_2a_ (PMab-231), we cloned the heavy and light chains of PMab-231 [52] into the pCAG-Neo and pCAG-Ble vectors, respectively.

The vectors were transfected into BINDS-09 (fucosyltransferase 8-knockout ExpiCHO-S; http://www.med-tohoku-antibody.com/topics/001_paper_cell.htm, accessed on
20 April 2024) cells using the ExpiCHO Expression System (Thermo Fisher Scientific, Inc.) to produce the defucosylated mAbs. The H_2_Mab-250-mG_2a_, tras-mG_2a_, H_2_Mab-250-hG_1_, trastuzumab, and PMab-231 were purified using Ab-Capcher (Kagawa, Japan). After washing with PBS, the bound antibodies were eluted with an IgG elution buffer (Thermo Fisher Scientific Inc.) and immediately neutralized using 1 M Tris-HCl (pH 8.0). Finally, the eluates were concentrated using Amicon Ultra (Merck KGaA) and replaced with PBS. The purified mAbs were confirmed by SDS-PAGE in reduced and non-reduced conditions (Appendix A).

Normal human IgG was purchased from Sigma-Aldrich Corp. (St. Louis, MO, USA).

### 4.3. ADCC

The ADCC of H_2_Mab-250-hG_1_ and trastuzumab was measured as follows. Human NK cells were purchased from Takara Bio, Inc. (Shiga, Japan) and were used as effector cells. The NK cells were used in the following experiment immediately after thawing. We labeled the target cells (BT-474, SK-BR-3, CHO-K1, and CHO/HER2) using 10 µg/mL Calcein AM (Thermo Fisher Scientific, Inc.). The target cells were plated in 96-well plates (1 × 10^4^ cells/well) and mixed with the human NK cells (effector to target ratio, 50:1) and 100 μg/mL of H_2_Mab-250-hG_1_, trastuzumab or control human IgG. The calcein release was measured after a 4.5 h incubation, and the cytotoxicity (% lysis) was calculated as described previously [34].

The ADCC of H_2_Mab-250-mG_2a_ and tras-mG_2a_ was measured as follows. Effector cells were obtained from the spleen of female BALB/c nude mice (Jackson Laboratory Japan, Inc., Kanagawa, Japan). The Calcein AM-labeled target cells (CHO-K1 and CHO/HER2) using 10 µg/mL Calcein AM were plated in 96-well plates (1 × 10^4^ cells/well) and mixed with the effector cells (effector to target ratio, 50:1) with 100 μg/mL of H_2_Mab-250-mG_2a_ and tras-mG_2a_ or control mouse IgG_2a_. After a 4 h incubation at 37 °C, the calcein release into the medium was measured, and the cytotoxicity (% lysis) was calculated.

### 4.4. CDC

The calcein-labeled target cells (BT-474, SK-BR-3, CHO-K1, and CHO/HER2) were plated and mixed with rabbit complement (final dilution 1:10, or 1:15 [Figure 2E], Low-Tox-M Rabbit Complement; Cedarlane Laboratories, Hornby, ON, Canada) and the indicated concentration of H_2_Mab-250-hG_1_, trastuzumab or control human IgG. Following incubation for 4.5 h at 37 °C, the calcein released into the medium was measured, as described above. In the case of H_2_Mab-250-mG_2a_ and tras-mG_2a_ or control mouse IgG_2a_, we performed a 4 h incubation at 37 °C and the cytotoxicity (% lysis) was calculated.

### 4.5. Antitumor Activities of H_2_Mab-250-hG_1_, Trastuzumab, H_2_Mab-250-mG_2a_, and Tras-mG_2a_, in Tumor Xenograft Models

To examine the antitumor effect of H_2_Mab-250-hG_1_, trastuzumab, H_2_Mab-250-mG_2a_, and tras-mG_2a_, animal experiments were approved by the Institutional Committee for Experiments of the Institute of Microbial Chemistry (approval no. 2023-066 and 2023-074). We determined the body weight loss exceeding 25% and maximum tumor size exceeding 3000 mm^3^ as humane endpoints.

CHO/HER2 cells were suspended in BD Matrigel Matrix Growth Factor Reduced (BD Biosciences, San Jose, CA, USA). Then, BALB/c nude mice (Jackson Laboratory Japan, Kanagawa, Japan) were injected subcutaneously in the left flank with 100 μL of the suspension (5 × 10^6^ cells). On days 9 and 16 post-inoculation, 100 μg of H_2_Mab-250-mG_2a_ (n = 8), tras-mG_2a_ (n = 8), or control mouse IgG_2a_ (PMab-231; n = 8) in 100 µL PBS was intraperitoneally injected.

For evaluation of H_2_Mab-250-hG_1_ and trastuzumab, we injected the mice with 100 μg of H_2_Mab-250-hG_1_ (n = 8), trastuzumab (n = 8), or control human IgG (n = 8) in 100 μL of PBS through intraperitoneal injection on days 7, 14, and 21 post-inoculation. Furthermore, human NK cells (8.0 × 10^5^ cells, Takara Bio, Inc.) were injected near the tumors subcutaneously on same days as the mAbs injection.

The tumor volume was calculated using the following formula: volume = W^2^ × L/2, where W is the short diameter and L is the long diameter. All the mice were euthanized by cervical dislocation.

Statistical analyses were performed using GraphPad PRISM 6 (GraphPad Software, Inc., La Jolla, CA, USA).

### 4.6. Pharmacokinetics of H_2_Mab-250-hG_1_ and Trastuzumab

HER2 ectodomain [34] was immobilized on Nunc Maxisorp 96-well immunoplates (Thermo Fisher Scientific Inc.) at a concentration of 1 µg/mL for 30 min at 37 °C. After washing with PBS containing 0.05% (*v*/*v*) Tween 20 (PBST; Nacalai Tesque, Inc.), the wells were blocked with 1% (*w*/*v*) bovine serum albumin (BSA)-containing PBST for 30 min at 37 °C. To make a standard curve, the serially diluted H_2_Mab-250-hG_1_ and trastuzumab (0.00064–10 µg/mL) were added to each well, followed by peroxidase-conjugated anti-human Fc (1:3000 diluted; Sigma-Aldrich Corp.). The enzymatic reactions were conducted using ELISA POD Substrate TMB Kit (Nacalai Tesque, Inc.), followed by the measurement of the optical density at 655 nm, using an iMark microplate reader (Bio−Rad Laboratories, Inc., Berkeley, CA, USA). The standard curve was produced using GraphPad PRISM 6. H_2_Mab-250-hG_1_ and trastuzumab (100 µg/mouse, n = 3) were intraperitoneally injected and the serums were collected from day 0 (4 h after injection) to 10. The concentration of mAbs was determined as described above. The half-life of the mAbs was calculated as described previously [53].

## Figures and Tables

**Figure 1 ijms-25-08386-f001:**
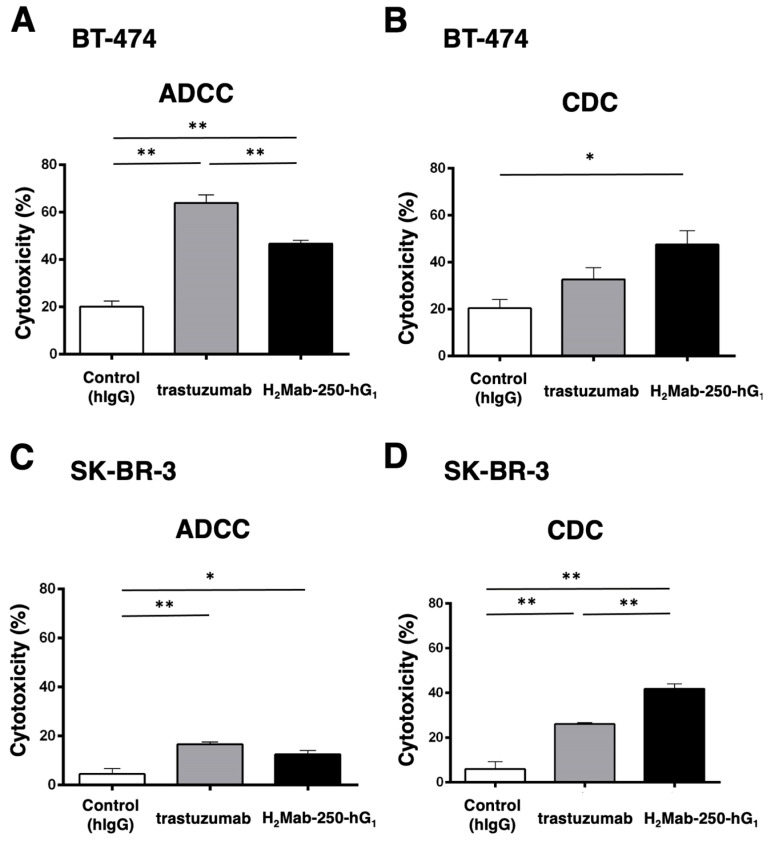
The ADCC and CDC activities were mediated by H_2_Mab-250-hG_1_ and trastuzumab in BT-474 and SK-BR-3 cells. (**A**,**C**) The ADCC induced by human NK cells in the presence of H_2_Mab-250-hG_1_, trastuzumab, or control human IgG (hIgG) against BT-474 (**A**) and SK-BR-3 (**C**) cells. (**B**,**D**) The CDC induced by complements in the presence of 100 µg/mL of H_2_Mab-250-hG_1_, trastuzumab, or control human IgG against BT-474 (**B**) and SK-BR-3 (**D**) cells. Values are shown as the mean ± SEM. Asterisks indicate statistical significance (** *p* < 0.01 and * *p* < 0.05; one-way ANOVA and Tukey’s multiple comparisons test).

**Figure 2 ijms-25-08386-f002:**
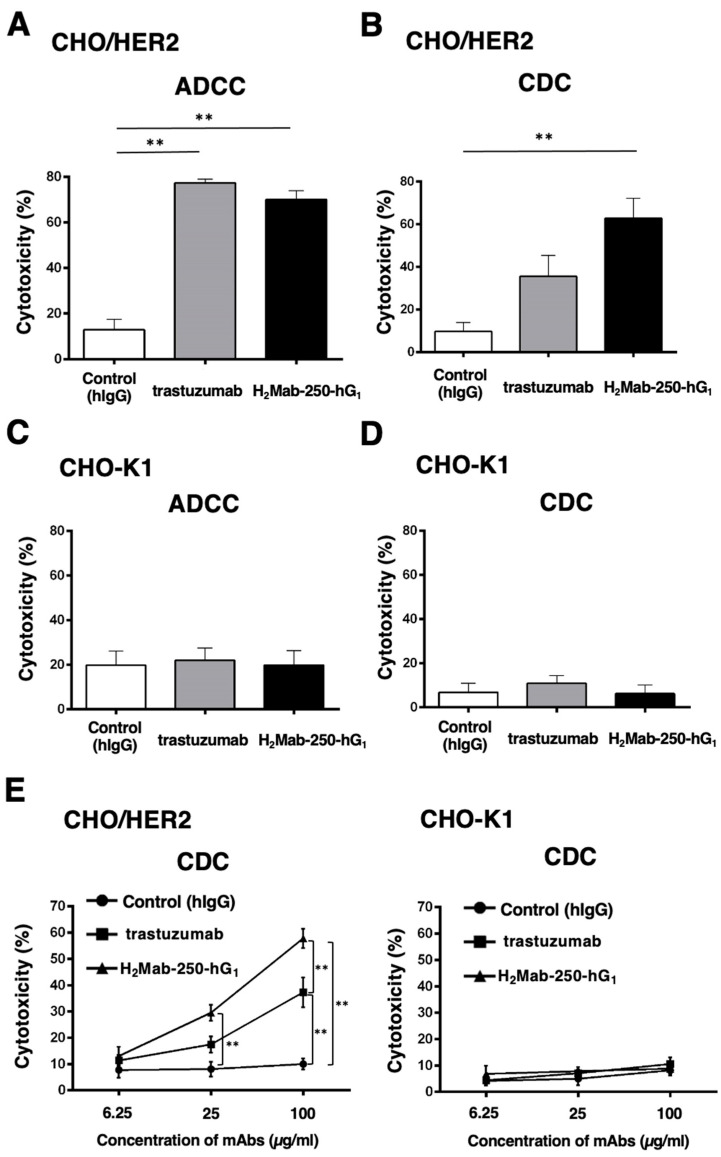
The ADCC and CDC activities are mediated by H_2_Mab-250-hG_1_ and trastuzumab in CHO/HER2 and CHO-K1 cells. (**A**,**C**) The ADCC induced by human NK cells in the presence of H_2_Mab-250-hG_1_, trastuzumab, or control human IgG (hIgG) against CHO/HER2 (**A**) and CHO-K1 (**C**) cells. (**B**,**D**) The CDC induced by complements in the presence of 100 µg/mL of H_2_Mab-250-hG_1_, trastuzumab, or control hIgG against CHO/HER2 (**B**) and CHO-K1 (**D**) cells. Values are shown as the mean ± SEM. Asterisks indicate statistical significance (** *p* < 0.01; one-way ANOVA and Tukey’s multiple comparisons test). (**E**) The CDC induced by complements in the presence of 6.25, 25, and 100 µg/mL of H_2_Mab-250-hG_1_, trastuzumab, or hIgG. Values are shown as the mean ± SEM. Asterisks indicate statistical significance (** *p* < 0.01; two-way ANOVA and Tukey’s multiple comparisons test).

**Figure 3 ijms-25-08386-f003:**
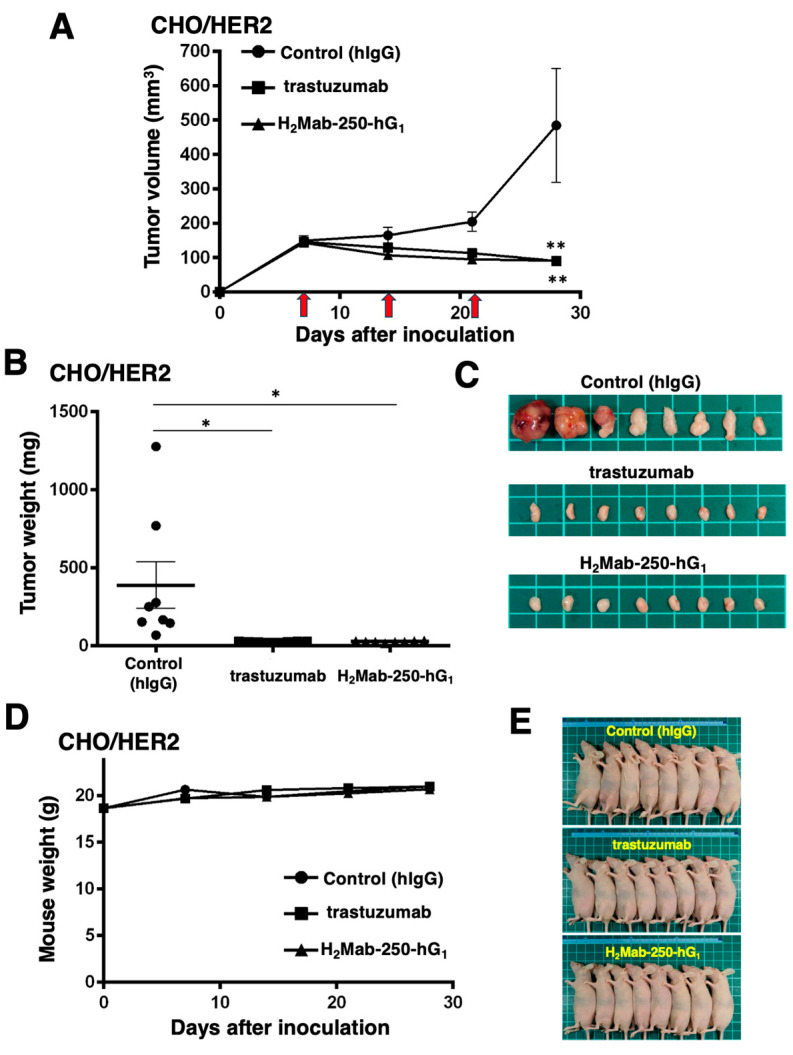
Antitumor activity of H_2_Mab-250-hG_1_ and trastuzumab against CHO/HER2 xenografts. (**A**) CHO/HER2 cells (5 × 10^6^ cells) were injected subcutaneously into the left flank of BALB/c nude mice (day 0). On day 7, 100 μg of H_2_Mab-250-hG_1_ (n = 8), trastuzumab (n = 8), or control human IgG (hIgG) (n = 8) was injected into the mice. On days 14 and 21, additional antibodies were injected. Human NK cells were injected around the tumors on the same days as the Ab administration (arrows). The tumor volume was measured on days 7, 14, 21, and 28. Values are presented as the mean ± SEM. ** *p* < 0.01 (two-way ANOVA and Tukey’s multiple comparisons test). The tumor weight (**B**) and appearance (**C**) of excised CHO/HER2 xenografts on day 28. Values are presented as the mean ± SEM. * *p* < 0.05 (two-way ANOVA and Tukey’s multiple comparisons test). The body weight (**D**) and appearance (**E**) of the xenograft-bearing mice treated with trastuzumab, H_2_Mab-250-hG_1_, or a control hIgG.

**Figure 4 ijms-25-08386-f004:**
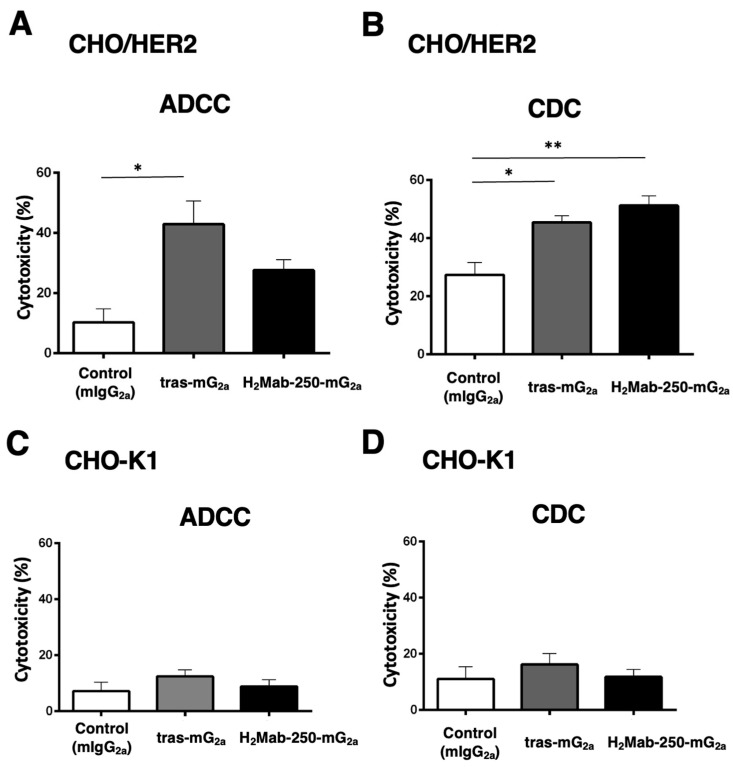
The ADCC and CDC activities are mediated by H_2_Mab-250-mG_2a_ and tras-mG_2a_ in CHO/HER2 and CHO-K1 cells. (**A**,**C**) The ADCC induced by human NK cells in the presence of 100 µg/mL of H_2_Mab-250-mG_2a_, tras-mG_2a,_ or control mouse IgG_2a_ (mIgG_2a_) against CHO/HER2 (**A**) and CHO-K1 (**C**) cells. (**B**,**D**) The CDC induced by complements in the presence of H_2_Mab-250-mG_2a_, tras-mG_2a,_ or control mIgG_2a_ against CHO/HER2 (**B**) and CHO-K1 (**D**) cells. Values are shown as the mean ± SEM. Asterisks indicate statistical significance (** *p* < 0.01 and * *p* < 0.05; one-way ANOVA and Tukey’s multiple comparisons test).

**Figure 5 ijms-25-08386-f005:**
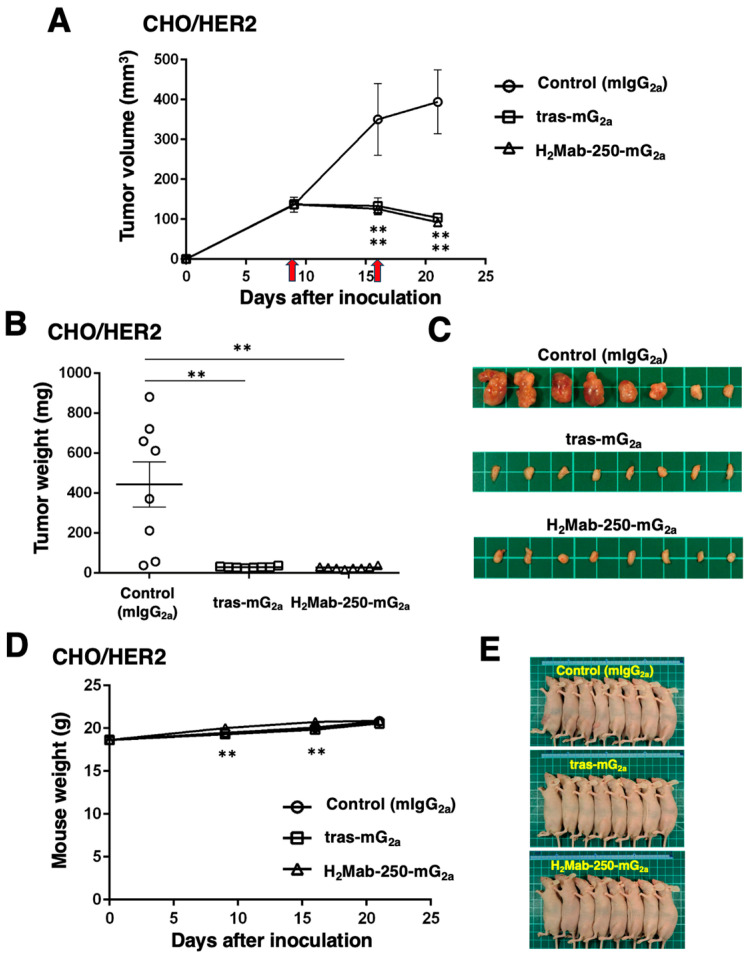
Antitumor activity of H_2_Mab-250-mG_2a_ and tras-mG_2a_ against CHO/HER2 xenografts. (**A**) CHO/HER2 cells (5 × 10^6^ cells) were injected subcutaneously into the left flank of BALB/c nude mice (day 0). On day 9, 100 μg of H_2_Mab-250-mG_2a_ (n = 8), tras-mG_2a_ (n = 8), or a control mouse IgG_2a_ (mIgG_2a_) (n = 8) was injected into mice. On day 16, additional antibodies were injected (arrows). The tumor volume was measured on days 9, 16, and 21. Values are presented as the mean ± SEM. ** *p* < 0.01 (two-way ANOVA and Tukey’s multiple comparisons test). The tumor weight (**B**) and appearance (**C**) of excised CHO/HER2 xenografts on day 21. Values are presented as the mean ± SEM. ** *p* < 0.01 (two-way ANOVA and Tukey’s multiple comparisons test). The body weight (**D**) and appearance (**E**) of xenograft-bearing mice treated with H_2_Mab-250-mG_2a_ and tras-mG_2a_, or a control mIgG_2a_.

**Figure 6 ijms-25-08386-f006:**
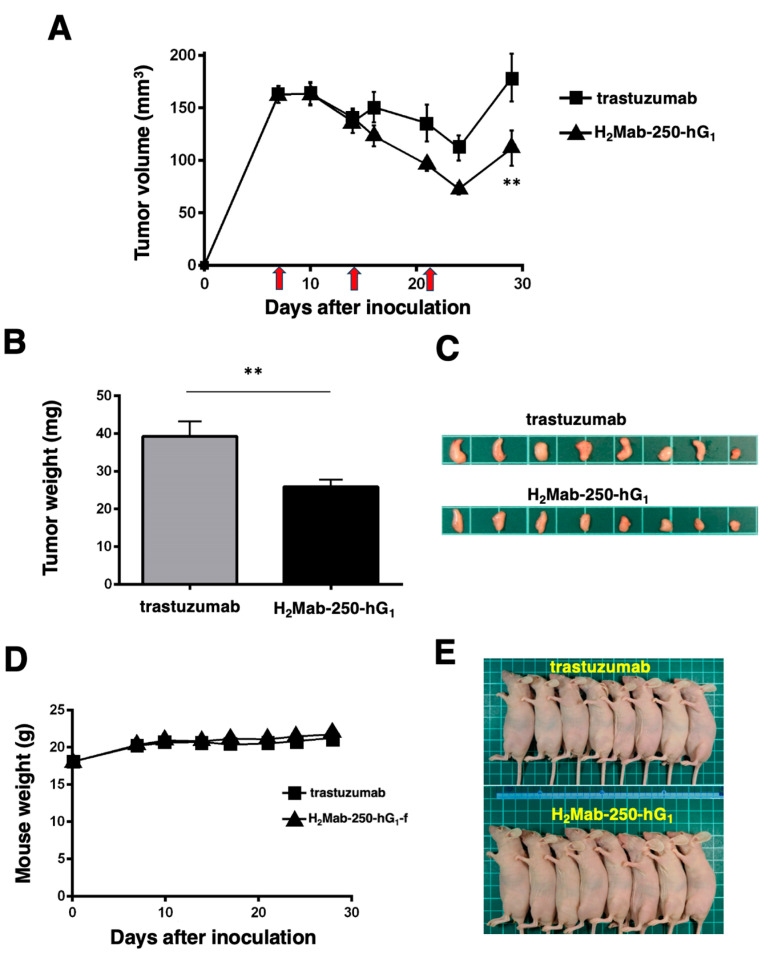
Antitumor activity of H_2_Mab-250-hG_1_ and trastuzumab against CHO/HER2 xenografts without human NK cell injection. (**A**) CHO/HER2 cells (5 × 10^6^ cells) were injected subcutaneously into the left flank of BALB/c nude mice (day 0). On day 7, 100 μg of H_2_Mab-250-hG_1_ (n = 8) or trastuzumab (n = 8) was injected into the mice. On days 14 and 21, additional antibodies were injected (arrows). The tumor volume was measured on days 7, 10, 14, 16, 21, 24, and 29. Values are presented as the mean ± SEM. ** *p* < 0.01 (two-way ANOVA and Tukey’s multiple comparisons test). The tumor weight (**B**) and appearance (**C**) of the excised CHO/HER2 xenografts on day 29. Values are presented as the mean ± SEM. ** *p* < 0.01 (two-way ANOVA and Tukey’s multiple comparisons test). The body weight (**D**) and appearance (**E**) of the xenograft-bearing mice treated with trastuzumab or H_2_Mab-250-hG_1_.

## Data Availability

The data presented in this study are available in the article.

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
