# Peer review of "Anti-HER2 Cancer-Specific mAb, H2Mab-250-hG1, Possesses Higher Complement-Dependent Cytotoxicity than Trastuzumab"

_ijms, 2024, doi:10.3390/ijms25158386_

Round 1
Reviewer 1 Report
Comments and Suggestions for Authors
In this manuscript Suzuki et al. describe the effect of H2Mab-250 with HER2-positive cancers in comparison with Trastuzumab, the mAB of election to treat these types of tumors.
The paper reported solid evidences about the efficacy of H2Mab-250 in terms of antibody-dependant cellular ciytotoxicity (ADCC) and complemement-dependant cellular cytotoxicity (CDC).
I would like to raise only one comment about graphics. Charts and images are very clear but I would suggest to reduce the font size, especially in the labels. They are very big a little bit invasise compared to the rest of the graphic.
Author Response
I would like to raise only one comment about graphics. Charts and images are very clear but I would suggest to reduce the font size, especially in the labels. They are very big a little bit invasise compared to the rest of the graphic.
>>> According to the comment, we changed the font size.
Reviewer 2 Report
Comments and Suggestions for Authors
In this paper, the authors measure in vitro ADCC and CDC activity of a new antibody developed by the authors and compare it to trastuzumab. In addition, the authors are evaluating the anti-tumor effects on mice transplanted with HER2-expressing CHO cells. However, there is no significant difference in content from the IJMS paper by the authors of reference 34 (Kaneko et al), and the data in this paper alone are far from sufficient to elucidate the anti-tumor mechanism of the title.
My comments are as follows;
1. To evaluate the in vivo efficacy of mAbs, it is important to consider pharmacokinetic (PK). (For example, Cancer Res (2010) 70 (11): 4481–4489.)To elucidate the mechanism, PK should be studied by imaging antibodies and/or measuring antibody levels in mice blood.
2. The authors have named H2Mab-250 as a cancer-specific anti-HER2 antibody and say it in line 149 and ref.34 also weakly binds to CHO/HER2. What does this mean? Isn't it specific to cancer HER2?
3. In the preparation of recombinant antibodies, you only mention that you used Ab-capture, but are you storing it as elution buffer?  Describe the buffer you used for storage and how it was stored until you used it in your experiment.
4. Concentration-dependent studies have not been performed for CDC activity, although ADCC activity has been done in Ref. 34. Data from only one concentration as high as 100 ug/mL is insufficient to compare the properties of the two antibodies.
5. The only mention of NK cells is that they were purchased from Takara Bio, but does that mean they were used in the experiment immediately after thawing? If not, you should describe the culture condition and preparation methods.
6. In line 388, you mention that NK cells were injected around the tumor. Is it between the tumor and the epithelium or is it subcutaneous near the tumor? You should be more specific about where you are injected it. And isn't it problematic as a comparative experiment to introduce it around the tumor when the tumor sizes are different? I think it should be administered intraperitoneally or by tail vein.
Reviewer 3 Report
Comments and Suggestions for Authors
The authors presented a research paper where a novel anti HER2 antibody has been compared with the oldest and already approved version “Trastuzumab” in order to elucidate the major immunology contribution to cancer treatment. The work is well written and well prepared. The Images are clear and well presented. Some revisions are suggested in order to improve the quality of the overall document.
1) Why H2Mab-250 was preferred to H2Mab-214? Please specify.
2) Would be nice to add a supplementary figure with the quality control of the used and produced antibodies (SDS-PAGE, Biacore, Size exclusion Chromatography)
3) In the figure in which is shown the tumor volume would be better to represent with an arrow the different injections, both antibodies and NK.
4) What’s happening after day 28 post-tumor implantation? Did you try to run an experiment to assess if a complete response occurred?
5) It’s preferable to represent the body weight not as it is but as a percentage of body weight change after treatments. For example in Figure 2 of this paper: doi: 10.1038/s44321-024-00034-0
6) Please specify the number of tumor cells injected and where also in figure captions.
7) Please give the immunological explanation for the NK injection that occurs when you use human antibodies and not for the mouse version.
8) Why did you change the schedule of immunization between the 2 in vivo experiments? 7-14-21 vs 9-16-21? It’s just a matter of weekends?
9) Please specify the type of H2Mab-250 antibody, fully human? Humanized? Chimeric? How was developed? Phage display? Hybridoma technology?
10) For M&M: Antibody purification method and production (Protein A?, Protein G?, storage buffer……)
11) Do you have in mind to test the possibility of the production of H2Mab-250-deruxtecan (T-DXd)?
Round 2
Reviewer 2 Report
Comments and Suggestions for Authors
Thank you for the revised manuscript. I think the authors have improved on the CDC data.
however, there is no significant difference in content from the paper by the authors of reference 33 and 34, and the lack of PK and mere addition of CDC data is not novel and far from elucidating the title anti-tumor mechanism.
Although the authors say that human chimeric antibodies are expected to have a short half-life, a few days is not short based on the authors' dosing schedule. 
Regarding CancerSpecific, the binding affinity of H2Mab-250, which recognizes misfolded HER2, has been studied against recombinant HER2ec in ref34. The antigen (misfolded HER2) that it recognizes is different from Trastuzumab, so the authors should do flowcytometry with different concentrations for CHO/HER2, for example, to produce binding affinity. Figure 1 of ref34 presented by the author in rebuttal, but I believe that both experiments were performed only at 10 ug and the binding curves are simply different. Wouldn't increasing the concentration bind to normal HER2?
Since basic data are lacking to begin with, the authors may be able to compare antitumor activity with trastuzumab, but the authors will be far from elucidating the mechanism.
Round 3
Reviewer 2 Report
Comments and Suggestions for Authors
The PK data revealed no significant difference between the authors' antibody and Trastuzumab.  If so, what do you think is the reason why the CDC activity of the authors' antibody is stronger than that of Trastuzumab?
This paper had a bit of novelty in its title of revealing the mechanism, but the new title loses that novelty as well. As I have said many times, if there is no mechanism elucidation, then it is no different than the previous paper. Nothing has been changed in the discussion and there is nothing novel about it.
The flow cytometry data (Supplementary Figure S1), using 2x concentration of antibody from ref34, shows a slight shift in results for HER2 expressed in normal cells, as I expected. This may not be cancer specific, but simply a low binding affinity to the HER2 antigen. It is meaningless without at least evaluating affinity for the antigen (misfolded HER2).
Author Response
The PK data revealed no significant difference between the authors' antibody and Trastuzumab.  If so, what do you think is the reason why the CDC activity of the authors' antibody is stronger than that of Trastuzumab?
This paper had a bit of novelty in its title of revealing the mechanism, but the new title loses that novelty as well. As I have said many times, if there is no mechanism elucidation, then it is no different than the previous paper. Nothing has been changed in the discussion and there is nothing novel about it.
Because the reviewer frequently asked the PK analysis, we had no chose but to do it.
Increased CDC activity was observed in vitro CDC assay (Fig. 1 and 2). At least, PK is not important in this assay. Therefore, we added the discussion as follows (line 323-).
Since ADCC mainly contributes the antitumor effect of anti-HER2 mAb including trastuzumab, the contribution of CDC of H2Mab-250 is thought to be important.
We added a discussion of the mechanism of CDC activity. The detailed molecular mechanism should be clarified by future investigation.
(line 323-) Furthermore, IgG antibodies can form ordered hexamers upon binding to their antigen on cell surfaces. These hexamers efficiently bind the hexavalent complement component C1q, the first step in the classical pathway of complement activation [39,40]. The structure of H2Mab-250-HER2 complex may provide the adequate access of complements to exert CDC. Further investigation and confirmation are required to clarify the mechanisms of CDC in H2Mab-250.The flow cytometry data (Supplementary Figure S1), using 2x concentration of antibody from ref34, shows a slight shift in results for HER2 expressed in normal cells, as I expected. This may not be cancer specific, but simply a low binding affinity to the HER2 antigen.
Serum concentration of mAbs were maintained around above concentration. Although a slight shift was observed, the difference is thought to be important.
It is meaningless without at least evaluating affinity for the antigen (misfolded HER2).
This is a newly added comment. This should be done at the first revision.